# Utilization and factors associated with health facility delivery among women of reproductive age in rural Ethiopia: Mixed effect logistic regression analysis

**Birhan Ewunu Semagn** *

Department of Public Health, Asrat Weldeyes Health Science Campus, Debre Berhan University, Debre Berhan, Ethiopia

* ewunubirhan@gmail.com

## Abstract

### Background

Worldwide over 800 women lose their life each day from complication in pregnancy and child birth. Health facility delivery is one of the key strategies for reducing maternal mortality and for ensuring safe birth. Inequity by urban–rural residence is one of the most pronounced challenges in maternal health service coverage with women living in rural areas at a greater disadvantage than other women. This study aims to assess the magnitude and factors affecting the utilization of health facility delivery for the most recent live birth among women of reproductive age in rural Ethiopia.

### Methods

This is a cross-sectional study based on a data from Ethiopian Mini Demographic and Health Survey 2019 dataset with a total weighted sample of 2900 women of reproductive age group in rural Ethiopia. Data cleaning, coding and labeling were done using STATA version 14 software. Multilevel mixed effect logistic regression model was employed to identify associated factors.

### Result

Only 44% of reproductive-age women in rural Ethiopia gave their most recent live birth in health institutions. In the multivariable multilevel binary logistic regression analysis; educational status, wealth index, attending 4+ANC, and had ANC from skilled provider were found to be statistically significant factors associated with health facility delivery.

### Conclusion

In a rural part of Ethiopia, the prevalence of institutional delivery is low. Especial emphasis should be given for mothers with no formal education, and poor household wealth index, Furthermore, implementing public health programs that target to enable women to have

**Data Availability Statement:** The data used for this study was EMDHS 2019 data, which is publicly available in the measure DHS program. The author accessed this data after explaining the purpose of

this study and therefore, everybody can access this data from this https://dhsprogram.com/data/ link.

**Funding:** The author received no specific funding for this work

**Competing interests:** The authors have declared that no competing interests exist.

**Abbreviations:** ANC, Antenatal Care; AOR, Adjusted Odds Ratio; CI, Confidence Interval; COR, Crude Odds Ratio; DHS, Demographic and Health Survey; EMDHS, Ethiopian Mini Demographic and Health Survey; SDG, Sustainable Development Goal.

more frequent Antenatal Care follow-up from skilled providers may increase the number of health facility deliveries.

## Background

Based on recent evidence there were decline in number of women and girls who lose their life each year related to complications of pregnancy and childbirth, with a decline from 451,000 in 2000 to 295,000 in 2017. But Still, we are losing over 800 women each day in death from complications in pregnancy and childbirth [1]. Despite all other reasons, low institutional delivery is one of the root causes of high maternal and newborn mortality [2].

Even though reducing global maternal mortality ratio (MMR) to lower than 70 per 100,000 live births is one of the Sustainable Development Goals(SDG) to be accomplished by 2030, maternal mortality mainly attributed to obstetric hemorrhage is still one of Africa's leading public health challenge [3, 4]. Despite there was good progress in reducing maternal mortality in Sub-Saharan African countries, there are the most off-track achievements of region-based maternal deaths, where the burden is still highest in rural women as compared to those urban women [5]. Over two thirds (68%) of all maternal deaths globally occurs in Sub-Saharan Africa with around 200,000 maternal deaths a year or 533 maternal deaths per 100,000 live births [1].

In low-income countries, most newborn deaths occur at home [6], and in rural Ethiopia, nearly one in every ten (11%) of neonates die before celebrating their first month of life, mainly during the first week [7]. Institutional delivery is one of the key strategies for reducing maternal mortality and for ensuring safe birth by reducing and intervening in any complications that will occur to the mother and her newborn during delivery and up to 24 hours postpartum [8, 9].

Even though addressing people who are more disadvantaged and have lower levels of health service utilization is one of the key parts of achieving SDG, inequalities by urban–rural residence are one of the most pronounced challenges in maternal health service coverage with women living in rural areas at a greater disadvantage than other women [10]. For achieving the 2030 development goal of health facility delivery in sub-Saharan Africa narrowing the gap or inequity between the rural and the urban areas is one of the ways forward [11]. Studies highlight that the urban-rural difference in institutional delivery was higher in East Africa especially the disparity is worst in the case of Ethiopia [12, 13]. Although the Federal Ministry of Health of Ethiopia initiated a free delivery service policy in all public health facilities to encourage mothers to deliver in health facilities, utilization of institutional deliveries remains minimal with a pooled prevalence of 31% [2, 14]. In Ethiopia, based on the most recent Ethiopian Mini Demographic and Health Survey (EMDHS) 2019 report seventy percent of live births in the 5 years before EMDHS2019 from urban women were delivered in a health facility while only forty percent of live births from rural women were delivered in a health facility [15]. Therefore, highlighting important factors for the designing and implementation of tailored public health interventions for improving institutional delivery in rural Ethiopia is needed.

Previous research has shown the magnitude and factors associated with institutional delivery in Ethiopia [14, 16–21], but as per the knowledge of the author, no study in Ethiopia investigates the determinants of health facility delivery of reproductive-age women in rural Ethiopia using nationally representative data. The very few studies conducted previously were either based on a small sample or a small segment of the population of rural Ethiopia. Therefor this study aimed to fill this gap by assessing the magnitude and factors affecting the utilization of

health facility delivery for the most recent live birth among women of reproductive age in rural Ethiopia using data from the most recent EMDHS.

## Methods

### Study design, data source, and setting

This is a cross-sectional study using data extracted from the latest EMDHS 2019. The data were obtained from the Demographic and Health survey (DHS) website (https://dhsprogram.com/ ) after submitting a request justifying the aim of the study. The 2019 EMDHS is the second EMDHS and the fifth DHS conducted in Ethiopia from March 21, 2019, to June 28, 2019. The survey was implemented based on a nationally representative sample that provided estimates for the urban and rural areas at the national and regional levels. 8,885 women of reproductive age (age 15–49) were interviewed from a nationally representative sample of 8,663 households [15]. Ethiopia is a country in the Horn of Africa with a total area of 1,100,000 km2 and lies between latitude 3˚ and 15˚ north and longitude 33˚ and 48 east [22]. During the time of the survey (2019), Ethiopia had nine ethnic-based and politically autonomous regional states and two cities (Addis Ababa and Dire Dawa).

### Population and sampling procedure

This study used all women of childbearing age (15–49 years) with a live birth in the five years preceding the survey in rural Ethiopia. The most recent birth was considered for women with two or more live births during the five-year period. EMDHS 2019 used a two-step stratified cluster sampling method, in which sample households were selected in cluster enumeration areas (EAs). In the first stage, 305 EAs were selected (93 in urban areas and 212 in rural areas) with probability in proportion to EA size. In the second stage, a fixed number of 30 households in each cluster were selected. Further information related to the population, study area, data collection, sampling procedure, and questionnaires used in the survey were detailed in the 2019 EMDHS Report [15]. In the current analysis, as shown in the figure (Fig 1), a weighted total of 2900 mothers who resides in a rural part of Ethiopia were included.

### Study variables

An outcome variable is place of delivery, which is dichotomized as a "health facility" (if a woman gives birth in public, private, or NGO health institutions) and a "non-health facility" (if a woman gives birth either in home or any other places) [23].

The potential covariates considered to have an association with health facility delivery were chosen based on prior literature and based on the presence of the variable of interest in the 2019EDHS dataset [14, 19, 21, 24]. These variables were the woman's age, woman's educational status, wealth index, religion, household family size, sex of household head, mass media exposure, visiting skilled providers during Antenatal Care (ANC), history of giving birth to a boy or girl who was born alive but later died, frequency of ANC, and the timing of ANC.

### Description and measurement of independent variables

**Age of respondents.**   The age of the women was re-coded into three categories with values of "1" for 15–24, "2" for 25–34, and "0" for 35 and above.

**Educational status.**   This is the minimum educational level a woman achieved with a value of "0" for no education, "1" for primary education, "2" for secondary and higher education.

Household interviewed in 2019 EMDHS=8663

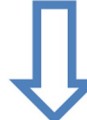

Weighted number of reproductive aged women interviewed in 2019 EMDHS =8885

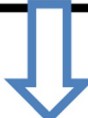

Weighted number of women interviewed about their most recent live birth in the last 5 years preceding the 2019 EMDHS =3927

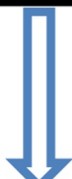

Weighted number of women in **rural** Ethiopia interviewed for their most recent live birth in the last 5 years preceding the 2019 EMDHS =2900

**Fig 1. The sampling procedure of study participants and the final sample size considered in this study from 2019 EMDHS dataset.**

**Wealth index.**    The datasets contained a wealth index that was created using principal components analysis coded as "poorest", "poorer", "Middle", "Richer", and "Richest in the EMDHS data set." For this study, recoded into three categories "poor" (includes the poorest and the poorer categories), "middle", and "rich" (includes the richer and the richest categories)

**Table 1. Model comparison between logistic regression and mixed effect logistic regression.**

| Proposed model | AIC value | BIC value | DIC value(-2*LL) |
|---|---|---|---|
| Logistic regression | 3076.36 | 3172.13 | 3044.36 |
| Mixed effect logistic regression | 2864.85 | 2966.61 | 2830.85 |

**Marital status.** This was the marital status of women during the survey and recoded into three categories with a value of "0" for never in union,"1" for those married or living with partner, and "2" for those widowed, divorced and no longer living together/separated

**Religion.** The variable religion was recorded as Orthodox, Muslim, Protestant, Catholic and others.

**Sex of household head.** The variable sex of household head was recorded as male and female in the dataset and we used without change.

**Having a son or daughter died.** A composite variable obtained by combining if a woman has a son or daughter died with a value of "0" if a woman didn't have a son or daughter died, and "1" if a woman has a son or daughter died.

**Media exposure.** A composite variable obtained by combining whether there was a radio and /or TV in the respondent's household with a value of "0" if a woman didn't have either TV or Radio in her household and "1" if a woman has access to either of the media.

**Household family size.** The family size of the women's household re-coded into two categories with values of "0" for a family size greater than 5, and "1" for a family size of less than or equal to 5.

**Region.** Geopolitical features of regions were grouped in to three categories: Metropolitan for Harrar and Drie-Dawa, Large central for Amhara, Oromia, South Nations and nationalities and Tigray, and Small peripheral for Afar, Benishangule, Gambella, and Somalia.

**Frequency of ANC visit.** The number of ANC visits during pregnancy were categorized into two groups and recoded as 1 "yes "if a woman had greater than or equal to four ANC, and 0 "No "if a woman didn't have greater than or equal to four ANC visit for the most recent live birth.

**Timing of ANC visit.** The timing of ANC visits was categorized into two groups and recoded as 1 "yes "if a woman has ANC visit in the first trimester of her pregnancy to the most recent live birth, 0 "No "if a woman didn't have ANC visit in the first trimester of her pregnancy to the most recent live birth.

**ANC by skilled providers.** A composite variable recoded as 1 "yes" if a woman received care from skilled providers, such as doctors, nurses/midwives, health officers, and health extension workers, and 0 "no "if she didn't receive care from either of these professions during her pregnancy of the most recent live birth.

## Data management and analysis

After extracting the data from EMDHS 2019, further coding and descriptive analysis were done using STATA version 14. The data was weighted using sampling weight, primary sampling unit, and strata before any statistical analysis to restore the representativeness of the survey and to tell the STATA to take into account the sampling design when calculating standard errors to get reliable statistical estimates. Due to the hierarchical nature of EMDHS data, women within the same cluster may be more similar to each other than women in the rest of the country. This violates the assumption of independence of observations and equal variance across clusters. This implies the need to use advanced models considering the between-cluster variability. Due to the dichotomous nature of the outcome variable logistic regression and

mixed effect Logistic regression were fitted. Model comparison was done using Akaike's information criterion (AIC) value, Bayesian information criterion (BIC) value ,and Deviance Information Criteria (DIC) [25]. A Mixed-effect model with the lowest AIC, BIC, and DIC were chosen (Table 1).

Furthermore, the Intra-cluster Correlation Coefficient (ICC) value was 0.47 which is in support of choosing mixed effect logistic regression over the basic model. Variables with p-values ≤0.2 in the bi-variable analysis were fitted in the multivariable model to measure the effect of each variable after adjusting for the effect of other variables. Adjusted Odds Ratio (AOR) with a 95% Confidence Interval (CI) and p-value < 0.05 in the multivariable model was declared as determinant factors associated with health facility delivery for the most recent live birth among women aged 15–49 in rural Ethiopia who had a live birth in the 5 years preceding the 2019 EMDHS. Multi-collinearity was also checked using Variance inflation factor (VIF), and a value of 10 was used as cut off.

## Results

### Characteristics of study populations

This study includes a weighted number of 2900 reproductive aged women in rural Ethiopia, who gave birth in the last 5 years preceding the 2019 EMDHS, and was interviewed for their most recent live birth. The majority of the study participants (49.8%) were between the age group of 25–34, and most of them (58.96%) didn't have formal education. Furthermore, only (23.9%) of them had media exposure to (TV & radio). The household wealth quintiles of (52.54%) of women were poor and below. Regarding their marital status and religion most of them were married /living with partners (94.88%), and orthodox follower (36%) in religion. More than half (54%) of the participants had a household family size of more than 5 individuals, and around 89% of them were from households headed by males Table 2.

### The magnitude of health institution delivery, and frequency and timing of the study population's ANC visit for their most recent live birth

Only 44% [38.00, 50.15] of reproductive-age women in rural Ethiopia gave their most recent live birth in health institutions. Only 22.32% [19.61,25.28] of the respondents had started their ANC visit during the first trimester of the most recent pregnancy. Even though most of the participants 69.67% [64.61,74.29] had their ANC visits from skilled providers, the majority of them 62.52% [58.51, 66.37] didn't attend four or more ANC visit Fig 2.

### Factors associated with health facility delivery for the most recent live birth

Since their p-value was greater than 0.2 at bi-variable analysis, variables like the sex of the household head, and having a son or daughter died were excluded from multivariable analysis. In the multivariable multilevel binary logistic regression analysis; educational status, wealth index, attending 4+ANC, and had ANC from skilled provider were found to be statistically significant factors associated with health facility delivery for the most recent live birth among women of reproductive age in rural Ethiopia.

The odds of giving birth at a health facility for the most recent live birth among women of reproductive age in rural Ethiopia with the educational status of primary, and secondary and higher were 1.72 (AOR = 1.72, 95% CI: 1.35–2.20), and 3.73 (AOR = 3.73, 95% CI: 2.33–5.98) times higher than women of reproductive age with no formal education.

The probability of giving birth in health facilities increased as the household wealth index increased. The middle wealth quintiles were 1.53 (AOR = 1.53, 95% CI: 1.15–2.03times more

**Table 2. Percent distribution of women aged 15–49 in rural Ethiopia who had a live birth in the 5 years preceding the 2019 EMDHS by socio-demographic characteristics according to a place of delivery for the most recent live birth from March 21, 2019, to June 28, 2019.**

| Variables | Place of delivery for the most recent live birth | | | | | | Weighted N |
| --- | --- | --- | --- | --- | --- | --- | --- |
| | Non-health facility | | Health facility | | Total | | |
| | % | CI | % | CI | % | CI | |
| **Age category** | | | | | | | |
| 15–24 | 11.22 | [9.16,13.67] | 13.57 | [11.28,16.23] | 24.78 | [22.46,27.27] | 719 |
| 25–34 | 28.64 | [24.93,32.67] | 21.16 | [17.99,24.71] | 49.8 | [46.92,52.69] | 1,444 |
| > = 35 | 16.15 | [13.90,18.70] | 9.26 | [7.67,11.14] | 25.42 | [23.08,27.90] | 737 |
| Total | 56.01 | [49.85,62.00] | 43.99 | [38.00,50.15] | 100 | | 2,900 |
| **Educational status** | | | | | | | |
| no education | 39.34 | [34.48,44.42] | 19.62 | [16.68,22.93] | 58.96 | [54.67,63.12] | 1,710 |
| Primary | 15.63 | [13.02,18.64] | 18.53 | [15.36,22.18] | 34.16 | [30.92,37.55] | 991 |
| secondary and above | 1.05 | [0.63,1.73] | 5.83 | [4.44,7.64] | 6.88 | [5.39,8.75] | 200 |
| Total | 56.01 | [49.85,62.00] | 43.99 | [38.00,50.15] | 100 | | 2,900 |
| **Wealth-index** | | | | | | | |
| Poor | 36.3 | [30.28,42.80] | 16.24 | [13.01,20.08] | 52.54 | [46.33,58.67] | 1,524 |
| Middle | 12.46 | [9.96,15.48] | 11.73 | [9.36,14.59] | 24.18 | [20.62,28.14] | 701 |
| rich | 7.25 | [5.59,9.36] | 16.02 | [12.42,20.42] | 23.28 | [18.90,28.32] | 675 |
| Total | 56.01 | [49.85,62.00] | 43.99 | [38.00,50.15] | 100 | | 2,900 |
| **Marital status** | | | | | | | |
| never in union | 0.22 | [0.08,0.55] | 0.32 | [0.12,0.85] | 0.53 | [0.26,1.07] | 15 |
| married/living with partner | 53.17 | [47.23,59.02] | 41.71 | [36.03,47.62] | 94.88 | [93.46,96.00] | 2,752 |
| divorced/no longer living together | 2.63 | [1.99,3.48] | 1.96 | [1.20,3.18] | 4.59 | [3.54,5.93] | 133 |
| Total | 56.01 | [49.85,62.00] | 43.99 | [38.00,50.15] | 100 | | 2,900 |
| **Religion** | | | | | | | |
| Orthodox | 18.01 | [14.55,22.09] | 18.01 | [14.43,22.25] | 36.02 | [30.58,41.86] | 1,045 |
| Muslim | 21.19 | [15.44,28.36] | 14.53 | [9.61,21.38] | 35.72 | [27.69,44.65] | 1,036 |
| Protestant | 15.2 | [11.03,20.58] | 11.01 | [7.42,16.03] | 26.21 | [19.75,33.88] | 760 |
| Catholic and others | 1.62 | [0.59,4.36] | 0.43 | [0.16,1.18] | 2.05 | [0.82,5.02] | 59 |
| Total | 56.01 | [49.85,62.00] | 43.99 | [38.00,50.15] | 100 | | 2,900 |
| **Sex of household head** | | | | | | | |
| female | 5.92 | [4.62,7.55] | 4.63 | [3.64,5.88] | 10.55 | [8.82,12.57] | 306 |
| male | 50.1 | [44.18,56.01] | 39.35 | [33.86,45.13] | 89.45 | [87.43,91.18] | 2,594 |
| Total | 56.01 | [49.85,62.00] | 43.99 | [38.00,50.15] | 100 | | 2,900 |
| **Given birth to a boy or girl who was born alive but later died** | | | | | | | |
| No | 52.87 | [47.09,58.58] | 42.64 | [36.75,48.74] | 95.51 | [94.18,96.56] | 2,770 |
| Yes | 3.14 | [2.24,4.38] | 1.34 | [0.89,2.03] | 4.49 | [3.44,5.82] | 130 |
| Total | 56.01 | [49.85,62.00] | 43.99 | [38.00,50.15] | 100 | | 2,900 |
| **House hold family size** | | | | | | | |
| greater than 5 | 34.89 | [30.33,39.74] | 19.2 | [16.11,22.71] | 54.08 | [50.42,57.70] | 1,568 |
| less than or equal to five | 21.13 | [18.43,24.11] | 24.79 | [20.93,29.09] | 45.92 | [42.30,49.58] | 1,332 |
| Total | 56.01 | [49.85,62.00] | 43.99 | [38.00,50.15] | 100 | | 2,900 |
| **Geopolitical features of regions** | | | | | | | |
| Metropolitans | 0.27 | [0.21,0.34] | 0.3 | [0.18,0.48] | 0.57 | [0.45,0.71] | 16 |
| Small peripheral regions | 6.08 | [5.14,7.18] | 1.82 | [1.35,2.46] | 7.9 | [6.95,8.97] | 229 |
| large central regions | 49.66 | [43.54,55.80] | 41.87 | [35.95,48.03] | 91.53 | [90.44,92.50] | 2,655 |
| Total | 56.01 | [49.85,62.00] | 43.99 | [38.00,50.15] | 100 | | 2,900 |
| **Media exposure (radio & TV)** | | | | | | | |

*(Continued)*

**Table 2.** (Continued)

| Variables | Place of delivery for the most recent live birth | | | | | | Weighted N |
|---|---|---|---|---|---|---|---|
| | Non-health facility | | Health facility | | Total | | |
| | % | CI | % | CI | % | CI | |
| no media exposure | 45.94 | [40.68,51.29] | 30.16 | [25.80,34.91] | 76.1 | [72.97,78.97] | 2,182 |
| has media exposure | 10.32 | [8.27,12.80] | 13.58 | [10.76,17.00] | 23.9 | [21.03,27.03] | 685 |
| Total | 56.26 | [50.08,62.25] | 43.74 | [37.75,49.92] | 100 | | 2,868* |

** N = 2868 because 32 respondents were not dejure residents to be asked about their media exposure.

likely to give birth in a health facility than those in the poor wealth quintiles. The rich wealth quintiles were 2.77 (AOR = 2.77, 95% CI: 1.98–3.88) times more likely to deliver their most recent live birth in a health facility than those in the poor wealth quintiles.

Looking at the frequency of ANC visit women made for the most recent live birth in 5 years preceding the 2019 EMDHS, women who had more than four ANC visits had 1.90 (AOR = 1.90, 95% CI: 1.50–2.40), times higher odds of giving birth at a health facility as compared to their counterparts.

Mothers who had ANC visit from skilled provider for the most recent live birth were 5.29 times more likely to give birth in a health facility than women who didn't have an ANC visit from skilled provider (AOR = 5.29, 95% CI: 3.96–7.07) Table 3.

## Discussion

This study aimed to assess the magnitude and factors affecting the utilization of health facility delivery of the most recent live birth among women of reproductive age in rural Ethiopia

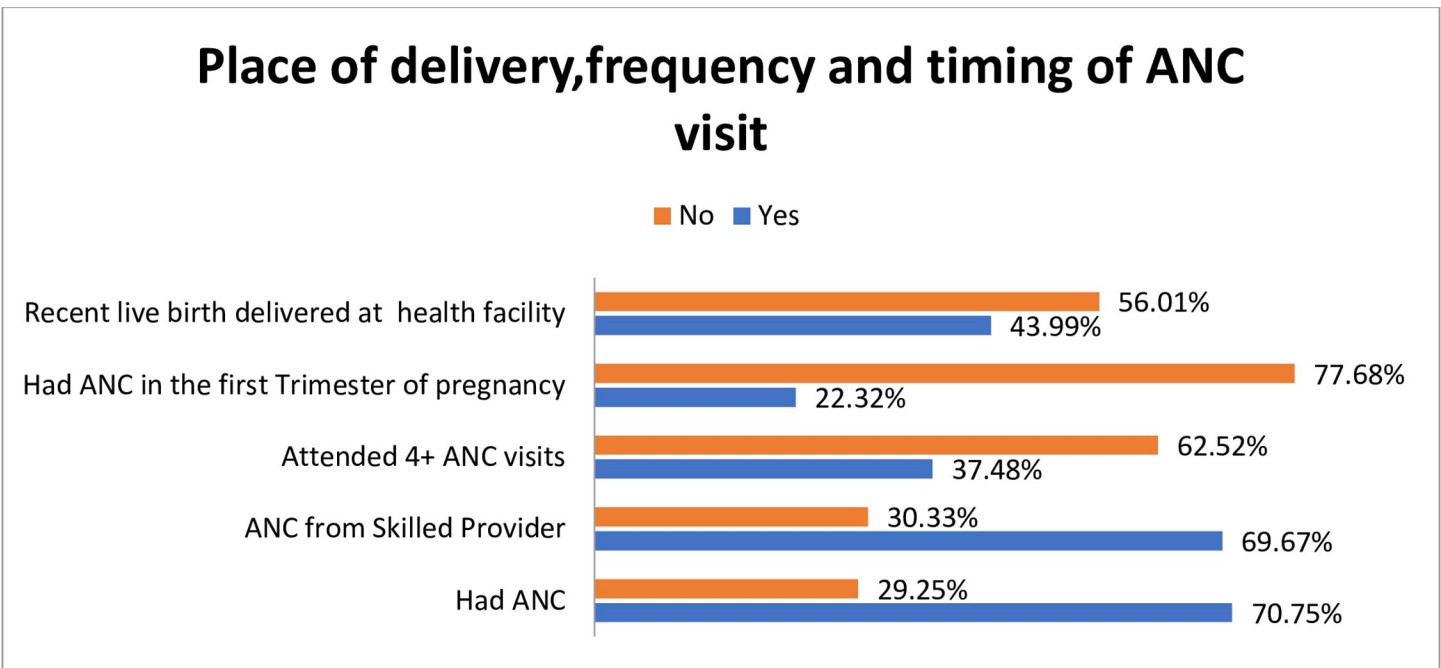

**Fig 2. Place of delivery, timing, and frequency of ANC visit for the most recent live birth among women of age 15–49 in rural Ethiopia who had a live birth in the 5 years preceding the 2019 EMDHS.**

**Table 3. Bivariable and multivariable multilevel binary logistic regression analysis of factors associated with health facility delivery for the most recent live birth among women aged 15–49 in rural Ethiopia who had a live birth in the 5 years preceding the 2019 EMDHS.**

| Variables | COR(CI) | | AOR (95%CI) | |
|---|---|---|---|---|
| **Age category** | | | | |
| 15–24 | 2.55** | 1.95–3.34 | 1.06 | 0.76–1.49 |
| 25–34 | 1.38** | 1.09–1.73 | 0.87 | 0.67–1.13 |
| > = 35 | 1 | | 1 | |
| **Educational status** | | | | |
| no education | 1 | | 1 | |
| primary | 2.29** | 1.85–2.83 | 1.72*** | 1.35–2.20 |
| secondary and above | 11.52** | 6.99–18.98 | 3.73*** | 2.33–5.98 |
| **Wealth-index** | | | | |
| Poor | 1 | | 1 | |
| Middle | 1.89** | 1.48–2.42 | 1.53*** | 1.15–2.03 |
| rich | 3.78** | 2.85–5.01 | 2.77*** | 1.98–3.88 |
| **Marital status** | | | | |
| never in union | 1 | | 1 | |
| married/living with partner | 0.61 | 0.17–2.22 | 0.40 | 0.12–1.33 |
| divorced/no longer living together | 0.40** | 0.10–1.55 | 0.43 | 0.12–1.52 |
| **Sex of household head** | | | | |
| Female | 1 | | | |
| Male | 1.08 | 0.78–1.48 | | |
| **Given birth to a boy or girl who was born alive but later died** | | | | |
| No | 1 | | | |
| Yes | 0.74 | 0.47–1.18 | | |
| **House hold family size** | | | | |
| greater than 5 | 1 | | 1 | |
| less than or equal to five | 1.96** | 1.62–2.38 | 1.23 | 0.98–1.54 |
| **Geopolitical features of regions** | | | | |
| Metropolitans | 1 | | 1 | |
| Small peripheral regions | 0.30** | 0.07–1.31 | 0.78 | 0.39–1.54 |
| large central regions | 1.01 | 0.24–4.13 | 0.92 | 0.47–1.78 |
| **Media exposure (radio & TV)** | | | | |
| no media exposure | 1 | | 1 | |
| has media exposure | 1.68** | 1.34–2.12 | 0.98 | 0.75–1.28 |
| **Attend 4+ANC visits** | | | | |
| No | 1 | | 1 | |
| Yes | 3.47** | 2.83–4.25 | 1.90*** | 1.50–2.40 |
| **Had ANC in the first trimester of pregnancy** | | | | |
| No | 1 | | 1 | |
| Yes | 2.59** | 2.06–3.27 | 1.27 | 0.99–1.63 |
| **Had ANC from skilled provider** | | | | |
| No | 1 | | 1 | |
| Yes | 7.18** | 5.50–9.39 | 5.29*** | 3.96–7.07 |

** p<0.2

***p<0.05

using data from the most recent EMDHS 2019. According to this study, only 44% of reproductive-age women in rural Ethiopia gave their most recent live birth in health institutions. This is consistence with a study conducted in different parts of Ethiopia [19, 26, 27], and rural Haiti [28].This magnitude of institutional delivery is lower than a study conducted in northwest Ethiopia [8, 29], women in rural Ghana [30–32], and rural women in Nepal [33], and it's higher than a study conducted in Nigeria [34]. This variation might be due to the difference in the study population in which a study conducted in rural Ghana was conducted among women who gave birth in the last 6 months of the data collection period while this study includes rural women in Ethiopia who gave birth in the last five year of the data collection period. Besides, the studies conducted in northwest Ethiopia were based on a small sample or small segment of a population of rural Ethiopia while the current study is based on representative data of the whole women of reproductive age in rural Ethiopia. And also, it might be due to the differences in socio-cultural characteristics as well as difference in utilization of maternal health services like ANC service. In this study majority of study populations 62.52% didn't attend four or more ANC visits for the most recent pregnancy whereas a study conducted in Ghana reports that 67.9%, and 75% of women attend four or more ANC visit during their recent pregnancy [31, 32].

In multivariable multilevel logistic regression analysis educational status, wealth index, , attending 4+ANC, and ANC from skilled provider were found to be statistically significant factors associated with health facility delivery for the most recent live birth among women of reproductive age in rural Ethiopia. Consistent with different studies conducted in Ethiopia [19, 35], Bangladesh [36], Ghana [31, 37], and Senegal [38] the probability of delivering in a health facility increases parallel with increasing women's educational status. Women with Primary, secondary and higher educational status had higher odds of giving birth in a health facility compared with women with no formal education. This might be because women with good educational status might have better information processing skills and improved cognitive skills that enable them to understand the purpose of health facility delivery and the risk of home delivery, which will result in the confidence to choose health facilities as a place of delivery [35]. Moreover, women with good educational status might have a high chance of reading and understanding information about health facility delivery [37].

In this study wealth index is another most important variable significantly associated with giving birth in a health facility for the most recent live birth among women of reproductive age in rural Ethiopia. That is women with middle and higher household wealth indexes were more likely to report institutional delivery as compared with women with poor household wealth indexes. This finding is consistent with a study conducted in Ethiopia [19, 35, 39], Uganda [40], India [41], Ghana [37, 42], and Cambodia [43]. Such discrepancy associated with wealth status might be due to the cost of transportation and any other extra cost associated with giving birth in health institutions [39]. In addition, women with poor household wealth status might have a low educational status that in turn affects their decision to give birth in health facility.

Moreover, in this study the frequency of ANC visit women made for the most recent live birth was significantly associated with the place of delivery, meaning that women who had more than four ANC visit had a higher probability of giving birth at a health facility as compared to their counterparts. This is in line with the previous study conducted in Ethiopia [8, 24, 35], and Ghana [31]. This might be due to the exposure of women with frequent ANC to repeated counseling about birth preparedness and complication readiness that can encourage mothers to deliver at a health facility [8].

Furthermore, mothers who had ANC visit from skilled provider for the most recent live birth were more likely to give birth in health facility than women who didn't have ANC visit from skilled providers. This might be due to the opportunity women got to have frequent

contact with health professionals that will enable them to get adequate information on the benefit of giving birth in a health facility to themselves and their newborn's health, as well as the women might acquire good awareness about the possible complications related with home delivery [32].

One of the strengths of this study was its trial to fill the gap of equity by addressing rural women by using large population-based data with large sample size, so it can be generalized to all women of reproductive age group in rural Ethiopia, and it will help as a baseline information to provide audience specific/tailored public health interventions in rural Ethiopia. Furthermore, the use of advanced statistical methods capable of accommodating the hierarchal nature of DHS data is also strength.

This study might have limitations. First, since we use secondary data some potentially important predictors were not available like distance from a health facility, knowledge, and attitude towards health facility delivery. Secondly, EMDHS 2019 was a questionnaire-based survey and asked women about their live births for the past five years before the survey, so recall bias might be the other limitation, but we try to minimize this by considering only the most recent live birth with in the past five years of the survey. Moreover, as this study is a cross-sectional study, it shares the limitation of cross-sectional study design. The author recommends more exploration using primary data to better understand the magnitude and determinants of institutional delivery among reproductive age group women in rural Ethiopia.

## Conclusion

In a rural part of Ethiopia, the prevalence of institutional delivery is low. Health facility delivery among reproductive age women of rural Ethiopia was significantly associated with educational status, wealth index, attending 4+ ANC, and having ANC visits from skilled providers. Thus, especial emphasis should be given to those mothers with no formal education, and poor household wealth index. Furthermore, implementing public health programs that target to enable women to have more frequent ANC follow-up from skilled providers may be an effective way to increase the number of health facility deliveries. Moreover, increasing the deployment of skilled healthcare professionals to rural Ethiopia might be effective in addressing the observed inequity.

## Supporting information

**S1 Checklist. STROBE statement—checklist of items that should be included in reports of observational studies.**
(DOCX)

## Acknowledgments

The author would like to extend his acknowledgment to the measure DHS for providing the data.

## Author Contributions

**Conceptualization:** Birhan Ewunu Semagn.

**Data curation:** Birhan Ewunu Semagn.

**Formal analysis:** Birhan Ewunu Semagn.

**Investigation:** Birhan Ewunu Semagn.

**Methodology:** Birhan Ewunu Semagn.

**Project administration:** Birhan Ewunu Semagn.

**Resources:** Birhan Ewunu Semagn.

**Software:** Birhan Ewunu Semagn.

**Validation:** Birhan Ewunu Semagn.

**Visualization:** Birhan Ewunu Semagn.

**Writing – original draft:** Birhan Ewunu Semagn.

**Writing – review & editing:** Birhan Ewunu Semagn.

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
