## [Decision Letter · Decision Letter 0]

24 May 2023

PONE-D-22-35754Utilization and Factors Associated with Health Facility Delivery among Women of Reproductive Age in Rural Ethiopia: Mixed effect logistic regression analysis.PLOS ONE

Dear Dr. Semagn,

Thank you for submitting your manuscript to PLOS ONE. After careful consideration, we feel that it has merit but does not fully meet PLOS ONE’s publication criteria as it currently stands. Therefore, we invite you to submit a revised version of the manuscript that addresses the points raised during the review process.

We look forward to receiving your revised manuscript.

Kind regards,

Anteneh Fikrie, MPH

Academic Editor

PLOS ONE

Journal Requirements:

**Additional Editor Comments:**

Result section under Table 3. Educational status: Higher (AOR=20.42; 95%CI:2.76 - 151.11) The confidence interval is too wide perhaps this could be due to the smaller sample size. Thus, I suggested the authors to consider merging Higher educational status and Secondary educational status so that you can create a new variable secondary and above will be created and the issue of wider confidence interval will be hopefully solved.Similarly, the authors considered Religion under statistical regression. What would be the implication if it was found to be statistically significant? 

Reviewers' comments:

Reviewer's Responses to Questions

**Comments to the Author**

1. Is the manuscript technically sound, and do the data support the conclusions?

Reviewer #1: Yes

2. Has the statistical analysis been performed appropriately and rigorously? 

Reviewer #1: Yes

3. Have the authors made all data underlying the findings in their manuscript fully available?

Reviewer #1: Yes

4. Is the manuscript presented in an intelligible fashion and written in standard English?

Reviewer #1: Yes

5. Review Comments to the Author

Reviewer #1: 1. Line 9 at the period of the study in 2022 , Ethiopia has 11 state and two cities ( the eleven state established as the South West Ethiopia Region was created on 23 November 2021).

2. Higher Educational status of AOR was too wide 20.42(2.76,151.11) it indicates that the precision of the estimate is low, meaning that there is a high level of uncertainty surrounding the true population parameter , thus it needs some correction .

3. Limitation of the study was not explained , so identifying and discussing the limitations, researchers can demonstrate their awareness of the scope and boundaries of their study, ensure transparency and objectivity in their reporting, and provide recommendations for future research to address the gaps or uncertainties.

6. PLOS authors have the option to publish the peer review history of their article (what does this mean?). If published, this will include your full peer review and any attached files.

Reviewer #1: **Yes: **Kebede Tefera (PhD, Assistant Professor in Public Health , Hawassa University )

---

## [Author Response · Author response to Decision Letter 0]

28 May 2023

Academic Editor: I have incorporated all of your suggestions in to my revision. Kindly look the file named “Response to reviewers”. They were very helpful. Thank you!

Reviewer 1: I have incorporated all of your suggestions in to my revision. Kindly look the file named “Response to reviewers”. They were very helpful. Thank you!

---

## [Editor Report · Decision Letter 1]

29 Jun 2023

Utilization and factors associated with health facility delivery among women of reproductive age in rural Ethiopia: Mixed effect logistic regression analysis.

PONE-D-22-35754R1

Dear Dr. **Ewunu B**,

We’re pleased to inform you that your manuscript has been judged scientifically suitable for publication and will be formally accepted for publication once it meets all outstanding technical requirements.

Kind regards,

Anteneh Fikrie, MPH

Academic Editor

PLOS ONE
---

## [Editor Report · Acceptance letter]

6 Jul 2023

PONE-D-22-35754R1 

Utilization and factors associated with health facility delivery among women of reproductive age in rural Ethiopia: Mixed effect logistic regression analysis. 

Dear Dr. Semagn:

I'm pleased to inform you that your manuscript has been deemed suitable for publication in PLOS ONE. Congratulations! Your manuscript is now with our production department. 

Kind regards, 

on behalf of

Professor Anteneh Fikrie 

Academic Editor

PLOS ONE